# Recognition of food images based on transfer learning and ensemble learning

**Le Bu, Caiping Hu**  *, **Xiuliang Zhang**

Department of Computer Engineering, Jinling Institute of Technology, Nanjing Jiangsu, China

* hucp@jit.edu.cn

## Abstract

The recognition of food images is of great significance for nutrition monitoring, food retrieval and food recommendation. However, the accuracy of recognition had not been high enough due to the complex background of food images and the characteristics of small inter-class differences and large intra-class differences. To solve these problems, this paper proposed a food image recognition method based on transfer learning and ensemble learning. Firstly, generic image features were extracted by using the convolutional neural network models (VGG19, ResNet50, MobileNet V2, AlexNet) pre-trained on the ImageNet dataset. Secondly, the 4 pre-trained models were transferred to the food image dataset for model fine-tuning. Finally, different basic learner combination strategies were adopted to establish the ensemble model and classify feature information. In this paper, several kinds of experiments were performed to compare the results of food image recognition between single models and ensemble models on food-11 dataset. The experimental results demonstrated that the accuracy of the ensemble model was the highest, reaching 96.88%, which was superior to any base learner. Therefore, the convolutional neural network model based on transfer learning and ensemble learning has strong learning ability and generalization ability, and it is feasible and practical to apply the method to food image recognition.

## 1. Introduction

Diet is an indispensable part of a person's day, and we need to take in nutrients from food to maintain the normal operation of the body, which plays a crucial role in human health and life safety [1–3]. Food is the source of human life, but it is also the root cause of diseases [4, 5]. With the increase of people's income level, the traditional diet has changed to a high-fat, high-calorie, low-fiber structure, which is one of the contributing factors to the high incidence of obesity [6]. Reasonable diet and balanced nutrition have become the pursuit goals of contemporary people. To help people develop healthy eating habits and get rid of obesity, automated analysis [7, 8] is expected to give people healthy diet recommendations [9, 10] in light of their daily food intake. Therefore, it is essential to explore a method that can both quickly classify the food images and detect the nutritional content of food.

Traditional machine learning methods [11–13] require manual definition of food image features, and it is inevitable that the extracted information is not comprehensive enough.

**Data Availability Statement:** This research employed publicly available datasets for its experimental studies. This is a dataset containing 16643 food images grouped in 11 major food categories. The 11 categories are Bread, Dairy product, Dessert, Egg, Fried food, Meat, Noodles/

Pasta, Rice, Seafood, Soup, and Vegetable/Fruit. Website https://www.kaggle.com/datasets/trolukovich/food11-image-dataset.

**Funding:** This research was funded by Jinling Institute of Technology High-level Talent Research Start-up Project (jit-rcyj-202102), Key R&D Plan Project of Jiangsu Province (BE2022077), Jiangsu Province College Student Innovation Training Program Project (202313573080Y, 202313573081Y) and Jinling Institute of Technology Science and Education Integration Project (2022KJRH18).The funders had no role in study design, data collection and analysis, decision to publish, or preparation of the manuscript.

**Competing interests:** The authors have declared that no competing interests exist.

However, food images themselves have the characteristics of small inter-class differences and large intra-class differences, so the recognition accuracy for food images had always been unsatisfactory. Deep learning, as an important branch of machine learning, has the ability to automatically learn, obtain high-dimensional features and analyze abundant internal information [14–16], reducing the workload of manually designing features. It has made breakthrough progress in many fields such as face recognition [17, 18], vehicle recognition [19], and speech recognition [20].

Since food image recognition is the basis of other food-related research work, scholars in China and abroad have conducted a lot of research on food image recognition in deep learning and achieved remarkable results. Liang Huagang et al. [21] adopted a cascaded structure of a three-level food feature extraction network to realize the transfer of features from global to local, and added a feature pyramid structure to each level of the network. Zhang Gang et al. [22] transferred the pre-trained network to a self-built small-scale food dataset for model fine-tuning, and combined it with Support Vector Machine (SVM) to output image classification results. Bian Jing et al. [23] proposed a Chinese food name and raw material identification method based on transfer learning and Inception V3 network to realize the identification of Chinese food names and raw materials. Peng Geng et al. [24] proposed a better backbone network and lightweight recognition and localization framework to extract fine-grained features from food images. Chen X et al. [25] proposed the Chinese food dataset ChineseFoodNet and fused ResNet152, DenseNet121, DenseNet169, DenseNet201, and VGG19-BN to generate a new deep convolutional neural network for classification. Liang Yanchao et al. [26] targeted 19 common foods, using Faster RCNN for calorie estimation to make the volume estimation result closer to the true value.

Deep learning methods [27] highly rely on data. To maximize the performance and achieve higher accuracy, it is of great necessity to have a large amount of data as support. At present, the development of the food image database has just started. Although data can be obtained, the food images are not manually labeled. Only a few data samples will lead to serious overfitting of the trained network model.

In order to extract high-level semantic features of images, overcome data bottlenecks, and achieve higher accuracy in food image recognition tasks, this paper proposed a food image recognition method based on transfer learning and ensemble learning. This method utilized the advantage of convolutional neural networks being able to automatically extract image features. The pre-trained convolutional neural network models (VGG19 [28], ResNet50 [29–31], MobileNet V2 [32], AlexNet [33]) on the ImageNet dataset were transferred to the food-11 dataset for model fine-tuning, and then ensemble models were established through different combination strategies of base learners. Combined with the experimental data, the performances of single models and ensemble models on the food image dataset would be compared in this paper. The method proposed in this paper had achieved good results, indicating the feasibility of the model transfer and combination strategies applied to food image recognition.

## 2. Basic principles

In this paper, convolutional neural network, transfer learning and ensemble learning were used to build the integrated transfer model. Convolutional neural network is a special artificial neural network, which uses a mathematical operation called convolution to replace the general Matrix multiplication in at least one of its layers. Transfer learning is one of the frontier research directions of machine learning. Its goal is to apply the knowledge or patterns learned in a certain field or task to different but related fields or problems. Ensemble learning improves the reliability and accuracy of prediction results by using a variety of machine learning models.

## 2.1. Convolutional neural network

Before the emergence of Convolutional Neural Network (CNN) [34–36], any digital image meant processing millions of parameters. Traditional machine learning methods would result in significant differences in image parameters, which had an impact on recognition accuracy. As a feedforward neural network, the convolution neural network has the characteristics of weight sharing and translation invariance, which can effectively reduce the amount of computation while retaining the main features of the original image, which is helpful to solve computer vision problems.

The basic structure of convolutional neural networks mainly includes the commonly used layers of convolutional neural networks, such as convolutional layer, pooling layer, fully connected layer, and classification layer. The convolutional layer is responsible for extracting local features in the image. The pooling layer is used to significantly reduce the parameter level. The fully connected layer combines the extracted features nonlinearly. The classification layer is used to output the final result.

## 2.2. Transfer learning

Training convolutional neural networks requires a large amount of data. Data acquisition requires manual annotation, which is time-consuming and laborious. The huge computational resources required for training the network are difficult for many ordinary people. Transfer learning is an effective way to solve the problem of small sample data sets.

In recent years, the transfer learning technology [37, 38] has developed vigorously. It enables models to learn knowledge from the source field, and then through the transfer of existing labeled data to unlabeled data, a deep learning model suitable for the target field is trained.

Convolutional neural networks trained on large-scale datasets already have the ability to extract image features. The pre-trained network is used as the basic model, and then the model is fine-tuned on other tasks. Continuously adjusting parameters in a small range during the training process can be suitable for the food image classification problem to be studied in this paper. Therefore, the advantage of transfer learning is that it does not need to train a model from scratch, which saves a lot of training time and computing resources, and can also achieve good performance on small sample data sets.

According to the classification of the content to be transferred, transfer learning can be divided into sample transfer, feature transfer and parameter transfer. Parameter transfer, which involves fixing a portion of parameters and adjusting the remaining parameters, is the most widely used method. The parameters of all feature layers in the pre-trained network are retained through parameter transfer, which is applied to feature extraction of food images. The extracted features are classified by their own classifiers, which plays a certain role in alleviating the overfitting phenomenon. However, directly using pre-trained networks often does not achieve the best results. Therefore, in this paper, we would fine-tune the model, freeze some feature layers of the network, and update the parameters of some feature layers and the final fully connected layer.

## 2.3. Ensemble learning

Due to the complex background and difficult feature extraction of food images, the feature extraction ability of a single learner is ultimately limited, making it impossible to achieve high-precision classification. Ensemble learning [39], that is, the machine learning algorithm of "learning from others' strengths", trains several basic learners, uses basic learners in combination with strategies to build an ensemble model, and finally forms a strong learner, which

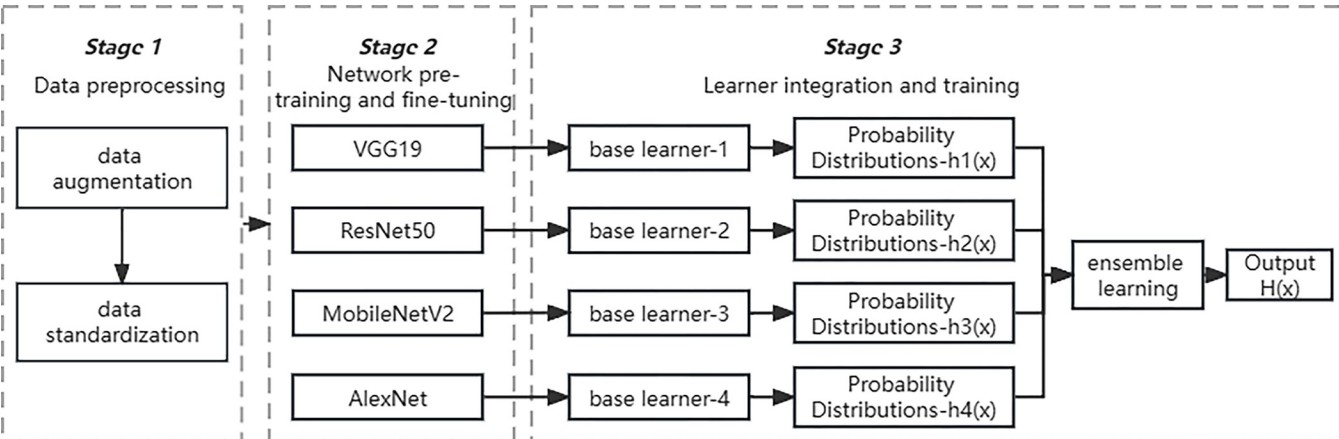

**Fig 1. The whole process of food image recognition based on transfer learning and ensemble learning.**

ensures the diversity of features extracted from the model, so as to obtain more stable, robust and accurate classification results, and can effectively restrain overfitting to a certain extent.

## 3. The recognition method for food images based on transfer learning and ensemble learning

The food image recognition method based on transfer learning and ensemble learning proposed in this paper mainly included three stages: data preprocessing, network pre-training and fine-tuning, learner integration and training. Fig 1 shows the overall flow chart of this method. In the data preprocessing stage, the preprocessing operation of food images was mainly divided into 2 steps: data augmentation and data standardization. In the stage of network pre-training and fine-tuning, four pre-trained convolutional neural networks (VGG19, ResNet50, MobileNet V2, AlexNet) on the ImageNet dataset were selected. Based on the extracted generic image features, the model was transferred to the food image dataset for fine-tuning training. Only a portion of the feature layers were frozen, and the remaining feature layers and all final fully connected layers were retrained. In the stage of learner integration and training, base learners were used to train and predict different feature vectors of the same image, and then different base learning combination strategies were used to establish an ensemble model. After further training and prediction, the final recognition result was obtained.

### 3.1. Data preprocessing

Data preprocessing is the foundation of analyzing digital images in machine vision. Its main purpose is to remove various noise in the image, improve the quality of the image, enhance the acquisition of useful information, simplify subsequent tasks such as feature extraction, image segmentation, and image recognition, and effectively improve the reliability of digital image analysis and processing. In this method, there were 2 steps for image preprocessing: data augmentation [40] and data standardization.

Data augmentation can effectively expand the sample size of the dataset, and sufficient training samples can enable convolutional neural networks to learn more diverse image feature information, ensuring that the model has strong generalization ability. The data augmentation strategies used in this paper mainly include: 1) randomly cropping; 2) horizontal flipping; 3) vertical flipping; 4) rotation with 180˚; 5) affine changes; 6) grayscale. Fig 2 shows a data augmentation example provided by this method.

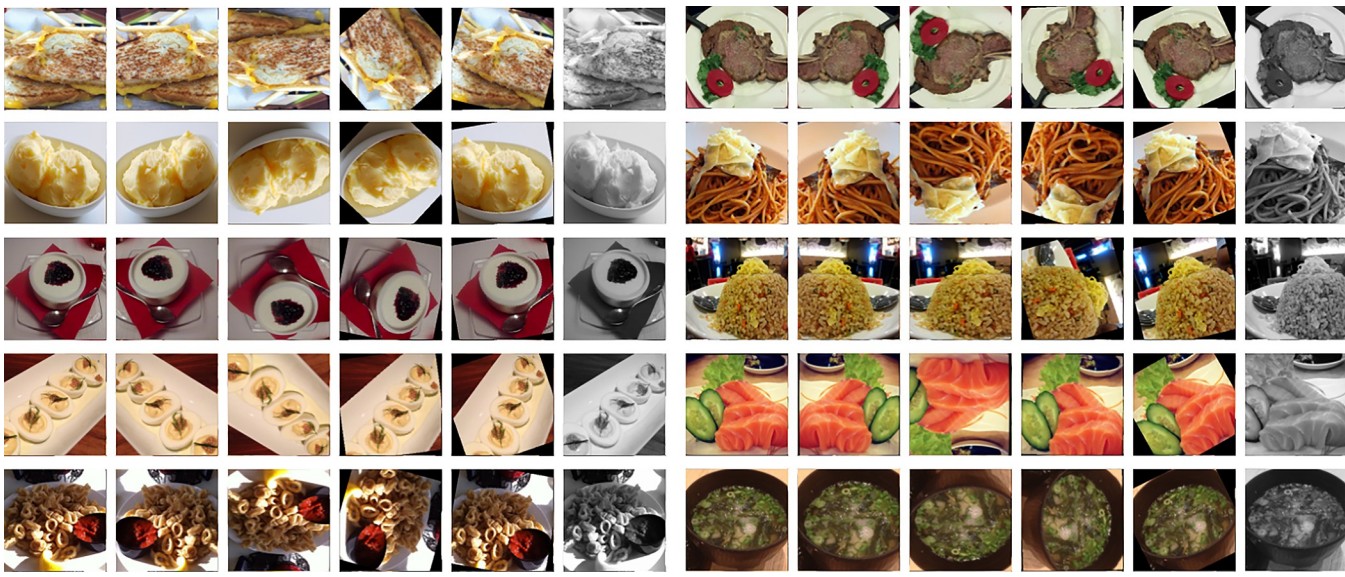

**Fig 2. Example of data augmentation.**

Data standardization is the standardization of each channel of an image, which subtracts the mean and then divides it by the standard deviation, which helps to converge the model during the training process.

## 3.2. Network pre-training and fine-tuning

ImageNet is one of the commonly used datasets in the field of computer vision, with 1.2 million training sets, 50,000 validation sets, and 100,000 test sets. The sample categories are mostly cats, dogs, vehicles, daily necessities, etc. Many researchers use it as a touchstone for deep learning algorithms and have an unshakable position in image classification, object segmentation, and object detection.

Transfer learning first required pre-training CNN on a large-scale ImageNet dataset. The four network structures selected in this paper, AlexNet, VGG19, MobileNet V2, and ResNet50, were all based on CNN and had certain representativeness. From AlexNet to ResNet50, the number of layers in the network continued to increase. VGG19 increased the nonlinear ability of the network, MobileNet V2 accelerated the computational speed of the model, and ResNet50 added cross layer connections to solve the problem of gradient vanishing. The depth of the network layer could be adjusted by increasing the number of convolutional layers, pooling layers, or fully connected layers. These four networks compensated for their respective shortcomings and facilitated the extraction of rich and complex high-dimensional image features, making them suitable for the food-11 dataset. Therefore, these four models were selected as base learners. Table 1 shows the basic parameters of each model and its performance comparison on the ImageNet dataset.

Due to the fact that the food image dataset is a small-scale dataset with low similarity to the ImageNet dataset, it is particularly important to retrain the network's feature layers. The insufficient size of the new dataset is compensated by freezing the first k layers of the pre-trained model, keeping the weights of the starting layer unchanged to obtain generic image features, and then fine-tuning the remaining feature layers and fully connected layers of the model to obtain new weights suitable for the new dataset.

**Table 1. Performance comparison of four convolutional neural networks on Imagenet.**

| Models | Layers | Size | Top-1/% | Top-5/% |
|---|---|---|---|---|
| VGG19 | 19 | 144 | 74.5 | 92.0 |
| ResNet50 | 50 | 25 | 75.3 | 93.3 |
| MobileNet V2 | 53 | 3.4 | 72.0 | 90.3 |
| AlexNet | 8 | 60 | 63.3 | 84.6 |

AlexNet is the first deep convolutional neural network applied to ImageNet dataset, and its accuracy has surpassed traditional methods. It consists of 5 convolutional layers and 3 fully connected layers. When fine-tuning the AlexNet model, the first 2 convolutional layers were frozen. The third convolutional layer and the last 3 fully connected layers were retrained.

VGG19 consists of 16 convolutional layers and 3 fully connected layers. Compared to Alex-Net, its depth has gradually increased and improvements have been made: using stacked 3x3 small convolutional kernels instead of large convolutional kernels. The advantage of using small convolutional kernels in VGG19 is that it can greatly reduce the number of parameters in the model while ensuring strong learning ability. When fine-tuning the VGG19 model, we retained the parameters of the first 7 convolutional layers. The last 6 convolutional layers and the last 3 fully connected layers were retrained.

As the number of layers in the network deepened, the accuracy of the network on the training set decreased instead of increasing, which was known as the "degradation" phenomenon of the network. In order to solve the problem of "degradation", ResNet50 introduced a residual structure, which made the network deepening model converge faster. ResNet50 consists of 49 convolutional layers and 1 fully connected layer. When fine-tuning the ResNet50 model, the parameters of the first 40 convolutional layers were retained. The last 9 convolutional layers and the last fully connected layer were retrained.

MobileNet V2 is a lightweight network based on the evolution of residual structure, mainly applied to mobile or embedded devices. MobileNet V2 consists of 53 convolutional layers and 1 fully connected layer. When fine-tuning the MobileNet V2 model, we retained the parameters of the first 42 convolutional layers. The last 11 convolutional layers and the last fully connected layer were retrained.

### 3.3. Learner integration and training

The commonly used strategies for combining base learners currently include the weight-ensemble method and the vote-ensemble method. Assuming there were T learners $\{h_1, h_2, h_3, \ldots, h_T\}$, for a certain prediction sample x, the prediction category is $\{c_1, c_2, \ldots, c_k\}$, and the prediction result of each learner is $\{h_1(x), h_2(x), \ldots, h_T(x)\}$.

The four base learners used in this paper were VGG19, ResNet50, MobileNet V2, and AlexNet. To further improve the recognition accuracy of the ensemble model, the prediction results of the four base learners on food images were first obtained, on this basis, the following 2 combination strategies would be adopted to establish the ensemble model:

(1) Weight-ensemble

When there is a significant difference in the performance of individual learners, the weight-ensemble method should be used. If each learner has a weight, the final prediction result is the weighted average of the individual learner's prediction results, i.e

$$H(x) = \sum_{i=1}^{T} w_i h_i(x) \tag{1}$$

$w_i$ represents the weight of the i-th base learner, and $w_i$ is calculated by $d_i$. The value of di is determined by the recognition accuracy of the base learner on the test set, sorted from low to high. The $d_i$ of the base learner with the highest recognition rate is 4, and the $d_i$ of the base learner with the lowest recognition rate is 1.

$$w_i = \frac{d_i}{1 + 2 + 3 + 4}, \sum_{i=1}^{T} w_i = 1 \tag{2}$$

(2) Vote-ensemble

The vote-ensemble method means that the minority submits to the majority. Among the prediction results of T learners on sample x, the category $c_i$ with the highest number is the final classification category. If multiple categories receive the highest vote at the same time, randomly select one from them.

## 4. Experimental results

### 4.1. Experimental environment and settings

Table 2 shows the experiment environment. To verify the effectiveness of the method proposed in this paper, the food image dataset used in the experiment was the Food-11 dataset [41] from the Multimedia Signal Processing Group (MSPG) of the Swiss Federal Institute of Technology. This is a dataset containing 16643 food images grouped in 11 major food categories. The 11 categories are Bread, Dairy product, Dessert, Egg, Fried food, Meat, Noodles/Pasta, Rice, Seafood, Soup, and Vegetable/Fruit. These foods are common types in our daily lives and meet the various nutrients our bodies need. We used six strategies for data augmentation on this dataset.

Firstly, by using direct training and fine-tuning training methods to train the model, the performance of four single convolutional neural network models (VGG19, ResNet50, MobileNet V2, AlexNet) on the food-11 dataset were compared and analyzed through experiments. Then, four single models were combined with different strategies of basic learners to establish an ensemble model. Experiments were conducted to compare the recognition effects of the single model and the ensemble model.

Table 3 shows the hyperparameter settings for training four convolutional neural network models. In direct training mode, initialize using a Gaussian distribution with a standard deviation of 0.001. In fine-tuning training mode, use four convolutional neural network models pre-trained on the ImageNet dataset to initialize weights, freeze the starting layer weights of each model, keep the starting layer weights unchanged to obtain universal image features, and then perform fine-tuning training on the remaining feature layers and fully connected layers of the model to obtain new weights suitable for the food-11 dataset. Both training modes used

**Table 2. Experiment environment.**

|  | Title | Description |
|---|---|---|
| Hardware Environment | CPU | Intel Xeon E5-2630L v3 |
|  | Memory | 64.0GB |
|  | GPU | NVIDIA GeForce RTX 3090 |
| Software Environment | OS | Ubuntu 16.04 LTS 64位 |
|  | Programming Language | Python |
|  | Development Environment | Anaconda3+Python 3.7.13+CUDA 11.3.1+ CUDNN 8.7.0+Pytorch-GPU 1.12.1 |
|  | Third Party Libraries | Numpy 1.15.1+Matplotlib 2.2.3 |

**Table 3. Hyperparameter settings for training four convolutional neural networks.**

| Models | Fine-tune Strategies | Learning Rate | Batch Size | Epoch |
|---|---|---|---|---|
| VGG19 | conv11-16[a]fc1-fc3[b] | 1e-4 | 64 | 150 |
| ResNet50 | conv41-49fc | 1e-4 | 64 | 150 |
| MobileNet V2 | conv43-53fc | 1e-4 | 64 | 150 |
| AlexNet | conv3-5fc1-fc3 | 1e-4 | 64 | 150 |

[a]Convolutional layer

[b]Fully-Connected layer.

the Adam optimizer for network training. The initial learning rate was set to 0.0001, the training cycle was 150 times, and the batch size was 64.

Different network models had different effects on image recognition, so different fine-tuning strategies had been adopted for different models. When fine-tuning the VGG19 model, the parameters of the first 7 convolutional layers were retained. The last 6 convolutional layers and the last 3 fully connected layers were retrained. When fine-tuning the ResNet50 model, the parameters of the first 40 convolutional layers were retained. The last 9 convolutional layers and the last fully connected layer were retrained. When fine-tuning the MobileNet V2 model, the parameters of the first 42 convolutional layers were retained. The last 11 convolutional layers and the last fully connected layer were retrained. When fine-tuning the AlexNet model, the first 2 convolutional layers were frozen. The third convolutional layer and the last three fully connected layers were retrained.

## 4.2. Evaluation indicator

In deep learning, there are many evaluation indicators used to evaluate model performance for image recognition tasks. The evaluation indicators used in this experiment mainly include the focal loss of the model on the training set and the accuracy of the model on the test set.

(1) Focal loss of the model on the training set

Focal loss function (FL) [42] is a commonly used loss function in the field of target detection. To address issues such as imbalanced sample categories and difficulty in sample classification, Focal Loss was used to train the model in the experiments. Focal loss, by modifying the cross entropy loss function (CE), can make the model increase the weight of a small number of target categories in training, and increase the weight of samples with wrong classification, so as to alleviate the above problems and improve the recognition accuracy. The calculation formula for focal loss is:

$$FL(p_t) = -\alpha_t(1 - p_t)^\gamma \log(p_t) \tag{3}$$

$-\log(p_t)$ is the initial cross entropy loss function, $\alpha_t$ is the weight parameter between categories, $(1-p_t)^\gamma$ is the simple/difficult sample adjustment factor, and $\gamma$ is the focusing parameter.

(2) The accuracy of the model on the test set

Definition of accuracy: The ratio of the correct number of samples predicted by the model to the total number of samples for a given test set.

$$P = \frac{C}{M} * 100\% \tag{4}$$

C represents the correct number of predicted samples, and M represents the total number of samples in the test set.

(3) F1-score

F1-score is a measure of the accuracy of a classification model, which is the harmonic average of accuracy and recall. The calculation formula is:

$$F1 = \frac{2(\text{recall} \times \text{precision})}{(\text{recall} + \text{precision})} \tag{5}$$

$$\text{precision} = \frac{TP}{TP + FP} \tag{6}$$

$$\text{recall} = \frac{TP}{TP + FN} \tag{7}$$

TP: True Positive, which means that the actual input is a positive sample and the predicted result is also a positive sample. FP: False Positive, which means that the actual input is a negative sample and the predicted result is a positive sample. FN: False Negative, which means that the actual input is a positive sample and the predicted result is a negative sample. TN: True Negative, which means that the actual input is a negative sample and the predicted result is a negative sample.

(4) FPS(Frames Per Second)

The FPS indicator represents the number of images processed in one second. The larger the FPS value is, the faster the model's detection speed is, which can be used to measure the model's detection efficiency.

## 4.3. Transfer learning experiment

When using four models, VGG19, ResNet50, MobileNet V2, and AlexNet, for transfer training on the enhanced food-11 dataset, a portion of the convolutional layers were frozen, and the remaining convolutional layers and the final fully connected layers were fine-tuned for training.

Fig 3 shows the changes in loss values of the four models during fine-tuning training. As the training period increased, the loss curves on the test set showed a gradual downward trend. Fig 4 shows the changes in accuracy values of the four models during fine-tuning training.

As the training period increased, the accuracy curves on the test set showed a gradual upward trend. Table 4 shows the experimental results of the four models in the direct training mode and the fine-tuning training mode were compared. It could be seen that the use of transfer learning strategies had significantly improved the recognition accuracy of models.

## 4.4. Comparison of results before and after model integration

In the ensemble learning stage, the weight-ensemble method and the vote-ensemble method were used to integrate the basic learners generated by transfer learning training. As shown in Fig 5, the recognition accuracy of the integration model under different integration strategies is shown. Table 5 shows the recognition accuracy of the single model and the ensemble model on the test set. It can be seen that the accuracy of the ensemble model established with two different integration strategies is higher than that of each basic learner, which indicates that the ensemble learning strategy has a certain improvement effect on the recognition accuracy, and the recognition accuracy of the weighted average method is the highest, reaching 96.88%. The method proposed in this paper combined the advantages of multiple models, sacrificing partial detection speed to improve performance. The time complexity of the method proposed in this paper is O (n*m), where n represents the number of training samples and m represents dividing the training samples into m training batches within one epoch.

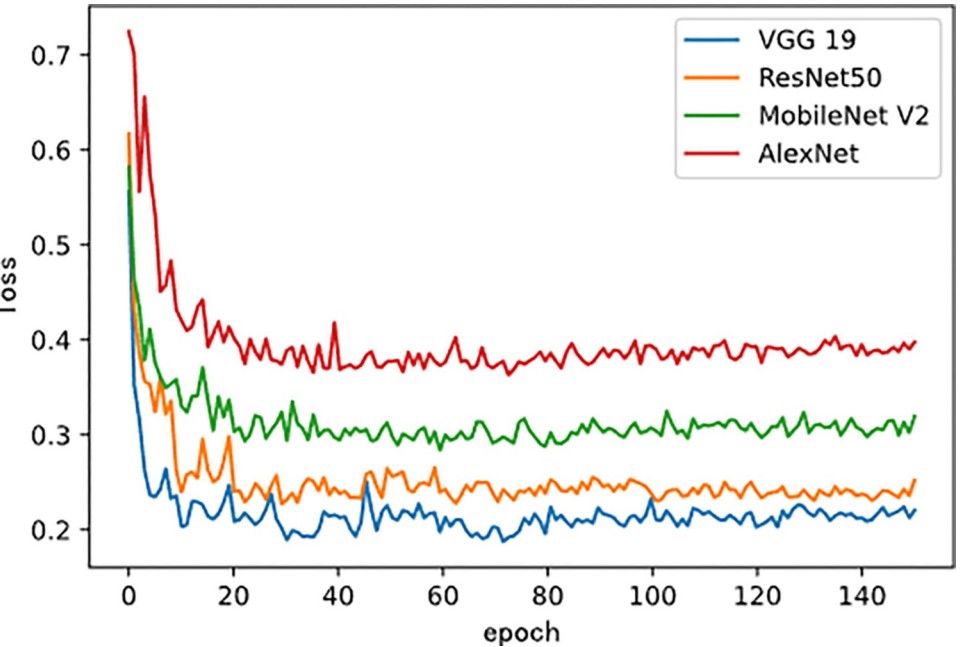

**Fig 3. Loss curves of four models during fine-tuning training.**

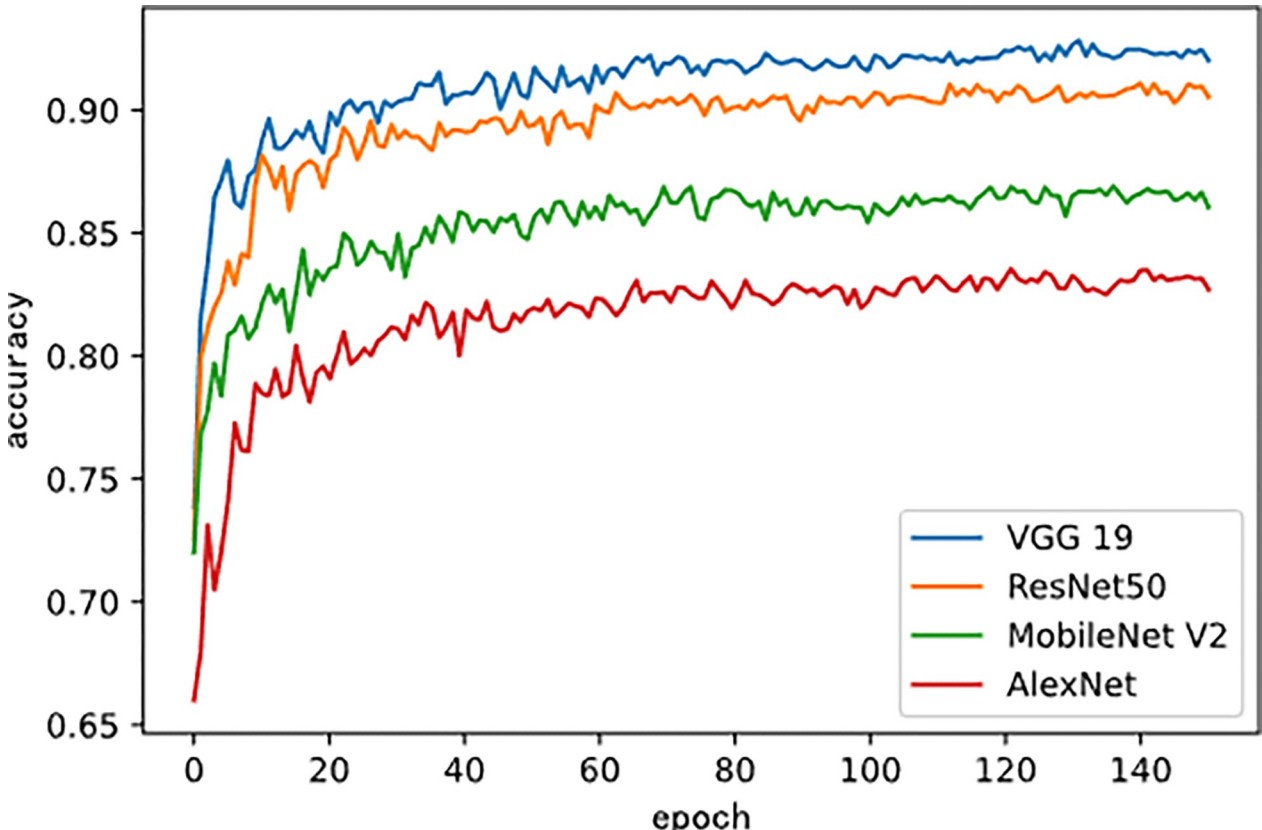

**Fig 4. Accuracy curves of four models during fine-tuning training.**

**Table 4. Comparison of experimental results of four convolutional neural networks in direct training mode and fine-tuning training mode.**

| Models | Accuracy(%) | Standard Deviation(%) | F1-score(%) |
|---|---|---|---|
| VGG19 | 81.32 | 8.53 | 80.28 |
| VGG19-ft[a] | 92.58 | 6.52 | 91.56 |
| ResNet50 | 80.56 | 8.43 | 79.35 |
| ResNet50-ft | 91.06 | 6.47 | 90.46 |
| MobileNet V2 | 73.69 | 8.69 | 72.48 |
| MobileNet V2-ft | 86.51 | 6.70 | 85.42 |
| AlexNet | 69.72 | 8.76 | 68.47 |
| AlexNet-ft | 83.25 | 6.92 | 82.16 |

[a]Fine-tuning training mode.

## 4.5. Comparison with the conventional methods

Prominent early image recognition techniques encompassed Support Vector Machines (SVM) and Backpropagation (BP) neural network, which enjoyed extensive utilization. Presently, improved convolutional neural networks are employed in image recognition methods to cater to diverse target recognition tasks. This experiment entailed training the model to recognize the food-11 dataset and the food-101 dataset using the image classification methods of SVM, BP neural network, and ResNet50 neural network. The HOG (Directional Gradient Histogram) method was used for feature extraction in the application of SVM and BP neural network, which improved the recognition performance. As shown in Table 6, the performance of these methods was compared with the proposed recognition approach based on transfer learning and ensemble learning.

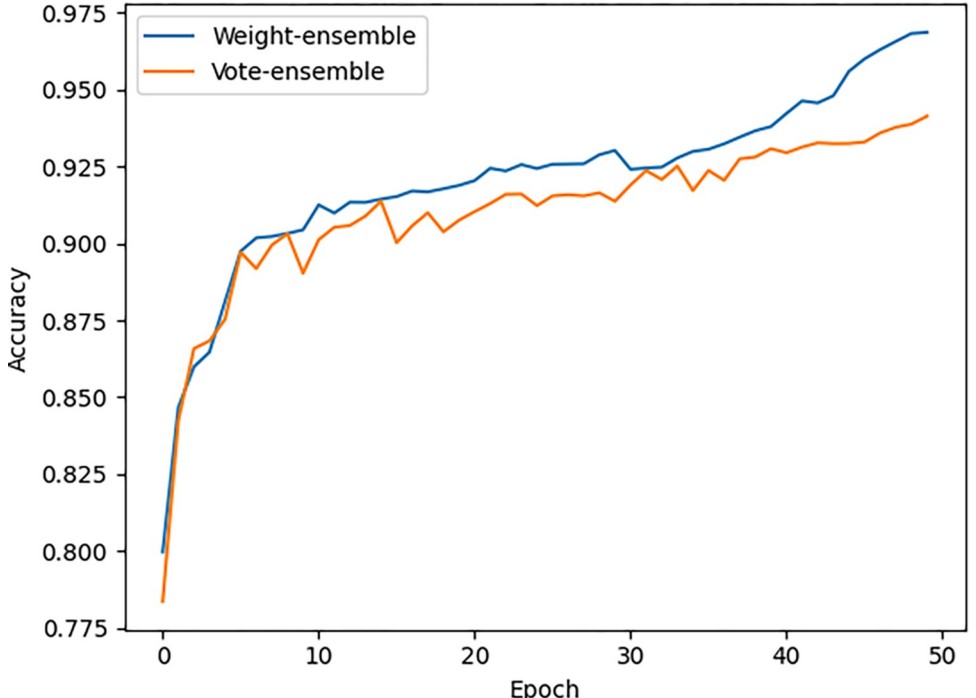

**Fig 5. Recognition accuracy of ensemble models under different ensemble strategies.**

**Table 5. Comparison of the results between single models and ensemble models.**

| Models | Accuracy(%) | F1-score(%) | FPS(f/s) |
|---|---|---|---|
| VGG19-ft[a] | 92.58 | 91.56 | 56.42 |
| ResNet50-ft | 91.06 | 90.46 | 73.56 |
| MobileNet V2-ft | 86.51 | 85.42 | 48.73 |
| AlexNet-ft | 83.25 | 82.16 | 120.58 |
| Weight-ensemble | 96.88 | 95.42 | 19.84 |
| Vote-ensemble | 94.37 | 93.18 | 19.25 |

[a]Fine-tuning training mode.

From the results as shown in Tables 6 and 7, it could be seen that the method based on transfer learning and ensemble learning had significantly improved recognition accuracy on the food-11 dataset and the food-101 dataset compared to the other methods. The main reason was that SVM and BP recognition methods mainly relied on manual feature extraction and continuous parameter adjustment for image recognition, and the recognition performance depended on prior knowledge and professional parameter adjustment experience. Although the ResNet50 method had improved recognition performance, it only applied the advantages of a single network. The method based on transfer learning and ensemble learning could automatically extract image features, integrate the advantages of various networks and improve the recognition performance of the model, thus having a relatively high recognition rate. Moreover, using the method proposed in this paper to train the food-101 dataset also achieved good results, indicating that the ensemble models based on transfer learning had good generalization performance.

## 5. Conclusions

In this paper, a food image recognition method based on ensemble learning and transfer learning was proposed to detect nutritional contents in common foods. Six strategies were used on food-11 dataset for data augmentation and a large number of comparative experiments were carried out for food recogniton. The results of four CNN models on the food-11 dataset

**Table 6. Comparison of the results between the proposed method and the conventional methods on the food-11 dataset.**

| Methods | Accuracy(%) |
|---|---|
| SVM+HOG | 72.54 |
| BP+HOG | 78.56 |
| ResNet50 | 88.24 |
| Ensemble models based on transfer learning | 96.88 |

**Table 7. Comparison of the results between the proposed method and the conventional methods on the food-101 dataset.**

| Methods | Accuracy(%) |
|---|---|
| SVM+HOG | 73.12 |
| BP+HOG | 79.20 |
| ResNet50 | 88.89 |
| Ensemble models based on transfer learning | 96.94 |

demonstrated that the transfer learning method could effectively reduce the overfitting phenomenon and improve the generalization performance of the model. By combining different learning strategies, it could be proven that the ensemble learning method effectively improved the accuracy of the model, and the weight-ensemble method was the best for ensemble learning, achieving an accuracy rate of 96.88%. This demonstrated that the use of deep convolution neural network could make full use of the high-level semantic features of images, and the recognition method based on ensemble learning and transfer learning could help the model learn more comprehensive and rich feature information, greatly improving the generalization performance of the model. Therefore, the method proposed in this paper is suitable for food image recognition under complex background.

## Author Contributions

**Conceptualization:** Caiping Hu.

**Investigation:** Le Bu.

**Project administration:** Xiuliang Zhang.

**Supervision:** Caiping Hu.

**Validation:** Xiuliang Zhang.

**Writing – original draft:** Le Bu.

**Writing – review & editing:** Caiping Hu.

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
