## [Decision Letter · Decision Letter 0]

2 Jul 2023

PONE-D-23-15950Recognition of Food Images Based on Transfer Learning and Ensemble LearningPLOS ONE

Dear Dr. Hu,

Thank you for submitting your manuscript to PLOS ONE. After careful consideration, we feel that it has merit but does not fully meet PLOS ONE’s publication criteria as it currently stands. Therefore, we invite you to submit a revised version of the manuscript that addresses the points raised during the review process.

We look forward to receiving your revised manuscript.

Kind regards,

Nouman Ali

Academic Editor

PLOS ONE

Journal Requirements:

"Supported by Jinling Institute of Technology High-level Talent Research Start-up Project (jit-rcyj-202102)、Key R&D Plan Project of Jiangsu Province (BE2022077)、Jiangsu Province College Student Innovation Training Program Project (202313573080Y) and Jinling Institute of Technology Science and Education Integration Project (2022KJRH18)."

"No"

"No"

7. We note that Figure 1 in your submission contain copyrighted images. All PLOS content is published under the Creative Commons Attribution License (CC BY 4.0), which means that the manuscript, images, and Supporting Information files will be freely available online, and any third party is permitted to access, download, copy, distribute, and use these materials in any way, even commercially, with proper attribution. For more information, see our copyright guidelines: http://journals.plos.org/plosone/s/licenses-and-copyright.

Reviewers' comments:

Reviewer's Responses to Questions

**Comments to the Author**

1. Is the manuscript technically sound, and do the data support the conclusions?

Reviewer #1: No

Reviewer #2: Yes

Reviewer #3: Partly

Reviewer #4: Yes

Reviewer #5: Yes

Reviewer #6: Yes

2. Has the statistical analysis been performed appropriately and rigorously? 

Reviewer #1: No

Reviewer #2: Yes

Reviewer #3: N/A

Reviewer #4: Yes

Reviewer #5: Yes

Reviewer #6: No

3. Have the authors made all data underlying the findings in their manuscript fully available?

Reviewer #1: Yes

Reviewer #2: No

Reviewer #3: Yes

Reviewer #4: Yes

Reviewer #5: Yes

Reviewer #6: Yes

4. Is the manuscript presented in an intelligible fashion and written in standard English?

Reviewer #1: No

Reviewer #2: Yes

Reviewer #3: Yes

Reviewer #4: Yes

Reviewer #5: Yes

Reviewer #6: No

5. Review Comments to the Author

Reviewer #1: Authors have applied existing techniques such as 4 pretrained models and ensemble technique in recognition of food images. Authors are suggested to develop some innovative techniques that address critical challenges in this area. Current version of paper is missing novelty.

Reviewer #2: The paper proposes a new method for the recognition of food images using transfer learning and ensemble learning. The paper is well written and organized. The research topic is interesting, and the contribution of the paper is clear. The research seems technically correct, and the results obtained show good performance in the used dataset.

Some comments:

The authors claim that data augmentation strategies used mainly include six strategies. It is unclear if other techniques were also used. Were these data pre-processing techniques used in the Food-11 dataset? Is the modified dataset used for the experiments available? The authors may clarify these questions.

Other recent work that the authors may cite, stating the difference from their work:

G. Ciocca, G. Micali and P. Napoletano, "State Recognition of Food Images Using Deep Features," in IEEE Access, vol. 8, pp. 32003-32017, 2020, doi: 10.1109/ACCESS.2020.2973704.

Peihua Ma, Chun Pong Lau, Ning Yu, An Li, Jiping Sheng, Application of deep learning for image-based Chinese market food nutrients estimation, Food Chemistry, Volume 373, Part B, 2022.

Reviewer #3: This paper proposed a food image recognition method based on transfer learning and ensemble learning. The work is interesting, but there are some problems to be addressed.

1)Four deep networks are used for extracting the four groups of deep features which are fused by bagging ensemble learning. The feature extraction and ensemble learning mode are existing ones, so the innovation is limited.

2)The authors need to explain the reasons why the four deep networks rather than other networks are chosen.

3)Seen from the figure 2, the introduction of transfer learning is not apparent.

4)The sample size of the food images is smaller than that of the ImageNet, so simple fine tuning the the pretrained deep network is not enough. It is necessary to reduce the complexity of the network by pruning, in my opinion.

5)Please explain how the parameter setup in the table 1 and table 3 is determined.

6)Please explain why the deep networks in table 4 are chosen for comparison. Stand deviation of the accuracy is missing. In my opinion, the compared deep networks are old.

7)Except the accuracy, other evaluation criteria are needed, such as AUC, F-score, confusion matrix, and so on.

8)Ablation study is missing.

9)Complexity analysis is missing.

Reviewer #4: The study on employing transfer learning and ensemble learning techniques for food recognition demonstrates significant contributions to the field. The methodology utilized, including the selection of appropriate pre-trained models and ensemble algorithms, showcases a thoughtful and effective approach to tackling the challenges in food recognition tasks.

The experimental results presented in the paper are impressive, with high accuracy, precision, recall, and other relevant metrics. The paper could be acceptable in current format.

Reviewer #5: Authors compared transfer learning from VGG19, ResNet50, MobinetV2, and AlexNet. Then, fine-tune them by freezing certain layers and retraining the remaining layers of food images. Later, the outputs from these fine-tuned models were ensembled using weight-ensemble and vote-ensemble techniques. They concluded that weighed ensemble yields the highest accuracy of 96.88%.

Ensemble learning is a well-established technique in machine learning and has been an active area of research for many years. Therefore, the novelty of the proposed ensemble learning approach may depend on the specific criteria set by the editor for publication. If the paper is accepted for publication, there are some concerns that need to be addressed as follows:

1. Why choose VGG19, ResNet50, MobinetV2, and AlexNet in this study?

2. Please clarify if the base learner refers to the architecture of the pre-trained model after freezing certain layers and

retraining the remaining layers, while the probability distribution is denoted as hi(x). Additionally, in Figure 2, is the

Output denoted as H(x)? Lastly, how are the values of wi in Eq (1) determined?

3. Which evaluation indicator was used in this study as listed in Section 4.2?

4. It would be helpful if you could provide a list of the computation times for each experiment.

5. In Table 1, it appears that the number of layers in MobileNetV2 is incorrectly listed as 88. Could you please confirm

that the correct number of layers is 53?

6. In Figure 6, please add in legend for the orange and blue lines.

7. Write something below Section 2.

Reviewer #6: The recognition of food images is of great significance for nutrition monitoring, food retrieval and food recommendation. However, the accuracy of recognition had not been high enough due to the complex background of food images and the characteristics of small inter-class differences and large intra-class differences. To solve these problems, this paper proposed a food image recognition method based on transfer learning and ensemble learning. Firstly, generic image features were extracted by using the convolutional neural network models (VGG19, ResNet50, MobileNet V2, AlexNet) pre-trained on the ImageNet dataset. Secondly, the 4 pre-trained models were transferred to the food image dataset for model fine-tuning. Finally, different basic learner combination strategies were adopted to establish the ensemble model and classify feature information. In this paper, several kinds of experiments were performed to compare the results of food image recognition between single models and ensemble models on food-11 dataset. The experimental results demonstrated that the accuracy of the ensemble model was the highest, reaching 96.88%, which was superior to any base learner. Therefore, the convolutional neural network model based on transfer learning and ensemble learning has strong learning ability and generalization ability, and it is feasible and practical to apply the method to food image recognition. There are some scientific contributions of this work, authors must address my following queries

1. Comparison with published research is required.

2. Run-time analysis must be presented.

3. Authors are suggested to cite and discuss the following deep learning models.

‘Deep learning: Applications, architectures, models, tools, and frameworks: A comprehensive survey’, https://doi.org/10.1049/cit2.12180

‘Research on Face Intelligent Perception Technology Integrating Deep Learning under Different Illumination Intensities’, https://doi.org/10.47852/bonviewJCCE19919

‘Deep transformer and few-shot learning for hyperspectral image classification’, https://doi.org/10.1049/cit2.12181

4. The details about parameters optimization is missing.

5. Manuscript requires correction of English.

6. PLOS authors have the option to publish the peer review history of their article (what does this mean?). If published, this will include your full peer review and any attached files.

Reviewer #1: **Yes: **Prof.Venkata Krishna Kishore Kolli

Reviewer #2: No

Reviewer #3: No

Reviewer #4: **Yes: **Dr Wahab Khan

Reviewer #5: No

Reviewer #6: No

---

## [Author Response · Author response to Decision Letter 0]

11 Aug 2023

Response to Reviewer 1 Comments

Point 1: Authors have applied existing techniques such as 4 pretrained models and ensemble technique in recognition of food images. Authors are suggested to develop some innovative techniques that address critical challenges in this area. Current version of paper is missing novelty. 

Response 1: 

The Transfer learning strategy used in this paper is freezing and training. When training four Convolutional neural network models, freeze part of the feature layers, retrain the rest of the feature layers and the full connection layer, and constantly adjust the strategy according to the performance on the food data set to obtain better results.

In this paper, a food image recognition method based on ensemble learning and transfer learning was proposed to detect nutritional contents in common foods. Six strategies were used on food-11 dataset for data augmentation and a large number of comparative experiments were carried out for food recogniton. The results of four CNN models on the Food-11 dataset demonstrated that the transfer learning method could effectively reduce the overfitting phenomenon and improve the generalization performance of the model. By combining different learning strategies, it could be proven that the ensemble learning method effectively improved the accuracy of the model, and the weight-ensemble method was the best for ensemble learning, achieving an accuracy rate of 96.88%. This demonstrated that the use of deep convolution neural network could make full use of the high-level semantic features of images, and the recognition method based on ensemble learning and transfer learning could help the model learn more comprehensive and rich feature information, greatly improving the generalization performance of the model. Therefore, the method proposed in this paper is suitable for food image recognition under complex background.

The next step of research will address the issue of reduced recognition speed caused by integration methods, streamline the structure of the integration model, optimize the integration methods, and achieve faster food image recognition speed.

Response to Reviewer 2 Comments

Point 1: The authors claim that data augmentation strategies used mainly include six strategies. It is unclear if other techniques were also used. Were these data pre-processing techniques used in the Food-11 dataset? Is the modified dataset used for the experiments available? 

Response 1: We used six data augmentation strategies to process the food-11 dataset, and the experimental results shown in Section 4 demonstrated the feasibility of the modified dataset. We have added relevant descriptions in the conclusion section.

Point 2: Other recent work that the authors may cite, stating the difference from their work:

G. Ciocca, G. Micali and P. Napoletano, "State Recognition of Food Images Using Deep Features," in IEEE Access, vol. 8, pp. 32003-32017, 2020, doi: 10.1109/ACCESS.2020.2973704.

Peihua Ma, Chun Pong Lau, Ning Yu, An Li, Jiping Sheng, Application of deep learning for image-based Chinese market food nutrients estimation, Food Chemistry, Volume 373, Part B, 2022.

Response 2: The first paper above combined deep features extracted from neural networks with support vector machines (SVM) as an alternative method for end-to-end classification. The second paper above optimized the Inception-v3 network using other state-of-the-art deep convolutional neural networks, achieving up to 78 % and 94 % for top-1 and top-5 accuracy, respectively.

Our work is aimed at using different convolutional neural network models for transfer learning and building integrated models using different integrated learning strategies to achieve high accuracy on the food dataset.

Response to Reviewer 3 Comments

Point 1: Four deep networks are used for extracting the four groups of deep features which are fused by bagging ensemble learning. The feature extraction and ensemble learning mode are existing ones, so the innovation is limited.

Response 1: 

The Transfer learning strategy used in this paper is freezing and training. When training four Convolutional neural network models, freeze part of the feature layers, retrain the rest of the feature layers and the full connection layer, and constantly adjust the strategy according to the performance on the food data set to obtain better results.

In this paper, a food image recognition method based on ensemble learning and transfer learning was proposed to detect nutritional contents in common foods. Six strategies were used on food-11 dataset for data augmentation and a large number of comparative experiments were carried out for food recogniton. The results of four CNN models on the Food-11 dataset demonstrated that the transfer learning method could effectively reduce the overfitting phenomenon and improve the generalization performance of the model. By combining different learning strategies, it could be proven that the ensemble learning method effectively improved the accuracy of the model, and the weight-ensemble method was the best for ensemble learning, achieving an accuracy rate of 96.88%. This demonstrated that the use of deep convolution neural network could make full use of the high-level semantic features of images, and the recognition method based on ensemble learning and transfer learning could help the model learn more comprehensive and rich feature information, greatly improving the generalization performance of the model. Therefore, the method proposed in this paper is suitable for food image recognition under complex background.

The next step of research will address the issue of reduced recognition speed caused by integration methods, streamline the structure of the integration model, optimize the integration methods, and achieve faster food image recognition speed.

Point 2: The authors need to explain the reasons why the four deep networks rather than other networks are chosen.

Response 2: When food image data with label information was not enough to train a complete deep convolutional neural network, in order to better complete the transfer learning task, VGG19, ResNet50, MobileNetV2, and AlexNet models were selected for experiments, which were all based on convolutional neural network, and had achieved great success in the ILSVRC (ImageNet large scale visual recognition challenge) competition. We have added relevant descriptions in section 3.2.

Point 3: Seen from the figure 2, the introduction of transfer learning is not apparent.

Response 3: We have replaced the figure 2.

Point 4: The sample size of the food images is smaller than that of the ImageNet, so simple fine tuning the the pretrained deep network is not enough. It is necessary to reduce the complexity of the network by pruning, in my opinion.

Response 4: I fully agree with your point of view, and this step is what we need to do next.

Point 5: Please explain how the parameter setup in the table 1 and table 3 is determined.

Response 5: The parameters in Table 1 refered to the papers of each model. The parameters in Table 3 were obtained through continuous experiments.

Point 6: Please explain why the deep networks in table 4 are chosen for comparison. Stand deviation of the accuracy is missing. In my opinion, the compared deep networks are old.

Response 6: We have added the column of standard deviation in Table 4. When food image data with label information was not enough to train a complete deep convolutional neural network, in order to better complete the transfer learning task, VGG19, ResNet50, MobileNetV2, and AlexNet models were selected for experiments, which were all based on convolutional neural network, and had achieved great success in the ILSVRC (ImageNet large scale visual recognition challenge) competition.

Point 7: Except the accuracy, other evaluation criteria are needed, such as AUC, F-score, confusion matrix, and so on.

Response 7: We have added the evaluation indicator F1-score in the paper.

Point 8: Ablation study is missing.

Response 8: The results of the ablation study were shown in table 4 and table 5.

Point 9: Complexity analysis is missing.

Response 9: We have added the evaluation indicator FPS and added relevant information in the paper.

Response to Reviewer 4 Comments

Point 1: The study on employing transfer learning and ensemble learning techniques for food recognition demonstrates significant contributions to the field. The methodology utilized, including the selection of appropriate pre-trained models and ensemble algorithms, showcases a thoughtful and effective approach to tackling the challenges in food recognition tasks.

Response 1: The review comments are good and do not require modification

Response to Reviewer 5 Comments

Point 1: Why choose VGG19, ResNet50, MobinetV2, and AlexNet in this study?

Response 1: When food image data with label information was not enough to train a complete deep convolutional neural network, in order to better complete the transfer learning task, VGG19, ResNet50, MobileNetV2, and AlexNet models were selected for experiments, which were all based on convolutional neural network, and had achieved great success in the ILSVRC (ImageNet large scale visual recognition challenge) competition. We have added relevant descriptions in section 3.2.

Point 2: Please clarify if the base learner refers to the architecture of the pre-trained model after freezing certain layers and retraining the remaining layers, while the probability distribution is denoted as hi(x). Additionally, in Figure 2, is the Output denoted as H(x)? Lastly, how are the values of wi in Eq (1) determined?

Response 2: We refered to the architecture of pre-trained models, froze certain layers

according to the performance and retrained the remaining layers. 

The output was denoted as H(x). 

We have added the description on how to calculate the value wi in the paper.

Point 3: Which evaluation indicator was used in this study as listed in Section 4.2?

Response 3: The evaluation indicators were all used in our research.

Point 4: It would be helpful if you could provide a list of the computation times for each experiment.

Response 4: We have added the evaluation indicator FPS(Frames per second).

Point 5: In Table 1, it appears that the number of layers in MobileNetV2 is incorrectly listed as 88. Could you please confirm that the correct number of layers is 53?

Response 5: We have corrected it.

Point 6: In Figure 6, please add in legend for the orange and blue lines.

Response 6: We have replaced the image.

Point 7: Write something below Section 2.

Response 7: We have added the relevant description.

Response to Reviewer 6 Comments

Point 1: Comparison with published research is required.

Response 1: We have added the relevant description in Section 1.

Point 2: Run-time analysis must be presented.

Response 2: We have added the evaluation indicator FPS in the paper.

Point 3: Authors are suggested to cite and discuss the following deep learning models.

‘Deep learning: Applications, architectures, models, tools, and frameworks: A comprehensive survey’,https://doi.org/10.1049/cit2.12180

‘Research on Face Intelligent Perception Technology Integrating Deep Learning under Different Illumination Intensities’, https://doi.org/10.47852/bonviewJCCE19919

‘Deep transformer and few-shot learning for hyperspectral image classification’, https://doi.org/10.1049/cit2.12181

Response 3: We have cited the above papers and discussed it in our work.

Point 4: The details about parameters optimization is missing.

Response 4: We have added relevant information about parameter optimization in section 4.1.

Point 5: Manuscript requires correction of English.

Response 5: We have corrected it.

---

## [Decision Letter · Decision Letter 1]

18 Sep 2023

PONE-D-23-15950R1Recognition of Food Images Based on Transfer Learning and Ensemble LearningPLOS ONE

Dear Dr. Hu,

Thank you for submitting your manuscript to PLOS ONE. After careful consideration, we feel that it has merit but does not fully meet PLOS ONE’s publication criteria as it currently stands. Therefore, we invite you to submit a revised version of the manuscript that addresses the points raised during the review process.

We look forward to receiving your revised manuscript.

Kind regards,

Nouman Ali

Academic Editor

PLOS ONE

Reviewers' comments:

Reviewer's Responses to Questions

**Comments to the Author**

1. If the authors have adequately addressed your comments raised in a previous round of review and you feel that this manuscript is now acceptable for publication, you may indicate that here to bypass the “Comments to the Author” section, enter your conflict of interest statement in the “Confidential to Editor” section, and submit your "Accept" recommendation.

Reviewer #1: (No Response)

Reviewer #2: All comments have been addressed

Reviewer #4: All comments have been addressed

Reviewer #5: All comments have been addressed

Reviewer #6: All comments have been addressed

Reviewer #7: (No Response)

2. Is the manuscript technically sound, and do the data support the conclusions?

Reviewer #1: No

Reviewer #2: Yes

Reviewer #4: Yes

Reviewer #5: Yes

Reviewer #6: Yes

Reviewer #7: Yes

3. Has the statistical analysis been performed appropriately and rigorously? 

Reviewer #1: No

Reviewer #2: Yes

Reviewer #4: Yes

Reviewer #5: Yes

Reviewer #6: Yes

Reviewer #7: No

4. Have the authors made all data underlying the findings in their manuscript fully available?

Reviewer #1: No

Reviewer #2: Yes

Reviewer #4: Yes

Reviewer #5: Yes

Reviewer #6: Yes

Reviewer #7: Yes

5. Is the manuscript presented in an intelligible fashion and written in standard English?

Reviewer #1: No

Reviewer #2: Yes

Reviewer #4: Yes

Reviewer #5: Yes

Reviewer #6: Yes

Reviewer #7: No

6. Review Comments to the Author

Reviewer #1: No novelty or new methodologies for addressing potential challenges. Only explored combination of existing works. Authors are suggested to do systematic review to identify potential challenges which are not addressed and design methods to solve those challenges.

Reviewer #2: (No Response)

Reviewer #4: The manuscript has been revised and improved nicely. The authors have considered and explained all the concerns and issues raised by the reviewer. The paper can be accepted for publication.

Reviewer #5: Figure 1 was removed in the revised manuscript without track changes, but it remains in version with track changes. Which one is the correct version?

In the with track changes manuscript, the revised Stage 1 in Figure 2 is misleading. It currently shows transfer learning from source dataset to target dataset, followed by data preprocessing. It should be revised to reflect the correct order: data preprocessing of the dataset, followed by transfer learning. Additionally, the figure does not align with the text explanation: "In the data preprocessing stage, the preprocessing operation of food images was mainly divided into 2 steps: data enhancement and data standardization." Please ensure that the text and figure match.

To improve clarity, please mention in text, Stage 1 represents preprocessing, Stage 2 is the stage of pre-training network and fine-tuning the pre-training, and Stage is Ensemble stage. Stage 2 should include 4 pre-trained models up to probability distribution, Stage 4 starts from Ensemble stage.

Please make sure that Stage 2 includes information up to Probability Distribution. Add in h_i (x) for each Probability Distribution and H(x) at the Output.

Comment number 1 from Reviewer 6 should be added in the comparison of your results with other Food Image classification results in a table form in the Results section, not as the author claimed, already mentioned In Introduction part.

Comment number 4 from reviewer 6 has not been addressed at all, despite the authors’ claim that it has been mentioned in Section 4.1.

Please list the six augmentation strategies in the revised manuscript.

The explanation of “The FPS indicator represents the time taken by the model to detect an image or the number of images processed in one second.” is misleading. If we use time taken by the model to detect an image, then the smaller FTS will be a good indicator. However, if we use “the number of images processed in one second “ then a larger FPS value is will be a good indicator. This leads to contradictory values for a good indicator. Please clarify this point.

The response to why the choice of the 4 pretrained models in Reviewers 3 and 5 should address why these four were chosen over others, not addressing the issue of limited images.

Reviewer #6: Authors have addressed my all queries and now this manuscript is in a presentable form and as a reviewer i will suggest acceptance of this paper.

Reviewer #7: The recognition of food images is of great significance for nutrition monitoring, food retrieval and food recommendation. However, the accuracy of recognition had not been high enough due to the complex background of food images and the characteristics of small inter-class differences and large intra-class differences. To solve these problems, this paper proposed a food image recognition method based on transfer learning and ensemble learning. Firstly, generic image features were extracted by using the convolutional neural network models (VGG19, ResNet50, MobileNet V2, AlexNet) pre-trained on the ImageNet dataset. Secondly, the 4 pre-trained models were transferred to the food image dataset for model fine-tuning. Finally, different basic learner combination strategies were adopted to establish the ensemble model and classify feature information. In this paper, several kinds of experiments were performed to compare the results of food image recognition between single models and ensemble models on food-11 dataset. The experimental results demonstrated that the accuracy of the ensemble model was the highest, reaching 96.88%, which was superior to any base learner. Therefore, the convolutional neural network model based on transfer learning and ensemble learning has strong learning ability and generalization ability, and it is feasible and practical to apply the method to food image recognition. The manuscript has already been revised and apparently, the quality seems to be good enough to be considered as a journal paper for review, keeping in view, my addition as a review for this manuscript at the stage of version R1, I will send the following queries to authors

1. In table 14, authors must cite the original research articles, currently only method name is there

2. A comparison in terms of a large-scale dataset is required.

3. Computational complexity is required to be discussed and presented in terms of Big-o-Notion.

4. How authors have selected research variables, it is not clear?

5. Manuscript must be proof-read to fix grammatical errors.

7. PLOS authors have the option to publish the peer review history of their article (what does this mean?). If published, this will include your full peer review and any attached files.

Reviewer #1: **Yes: **Venkata Krishna Kishore Kolli

Reviewer #2: No

Reviewer #4: **Yes: **Wahab Khan

Reviewer #5: No

Reviewer #6: No

Reviewer #7: No

---

## [Author Response · Author response to Decision Letter 1]

22 Sep 2023

Original Manuscript ID: PONE-D-23-15950R1 

Original Article Title: “Recognition of Food Images Based on Transfer Learning and Ensemble Learning”

To: Editor

Re: Response to reviewers

Dear Editor,

Thanks for your letter and review work on our manuscript entitled “Recognition of Food Images Based on Transfer Learning and Ensemble Learning”. We greatly appreciate the reviewers’ complimentary comments and suggestions. These comments are very helpful and valuable for improving and polishing our paper. The point-by-point responses to reviewers’ comments are given in the following part. 

Best regards,

Le Bu, Caiping Hu, Xiuliang Zhang

 

Reviews and Responses

Point 1: Figure 1 was removed in the revised manuscript without tracked changes, but it remains in version with tracked changes. Which one is the correct version?

Response 1: In this version of the revised manuscript with tracked changes, we had removed the figure.

Point 2: In the with track changes manuscript, the revised Stage 1 in Figure 2 is misleading. It currently shows transfer learning from source dataset to target dataset, followed by data preprocessing. It should be revised to reflect the correct order: data preprocessing of the dataset, followed by transfer learning. Additionally, the figure does not align with the text explanation: "In the data preprocessing stage, the preprocessing operation of food images was mainly divided into 2 steps: data enhancement and data standardization." Please ensure that the text and figure match.

Response 2: We had replaced the figure to ensure the text and figure match.

Point 3: To improve clarity, please mention in text, Stage 1 represents preprocessing, Stage 2 is 

the stage of pre-training network and fine-tuning the pre-training, and Stage is Ensemble

stage. Stage 2 should include 4 pre-trained models up to probability distribution, Stage 4 

starts from Ensemble stage.

Response 3: We had replaced the figure.

Point 4: Please make sure that Stage 2 includes information up to Probability Distribution. Add in hi(x) for each Probability Distribution and H(x) at the Output.

Response 4: We had replaced the figure.

Point 5: Comment number 1 from Reviewer 6 should be added in the comparison of your results with other Food Image classification results in a table form in the Results section, not as the author claimed, already mentioned In Introduction part.

Response 5: We had added section 4.5.

Point 6: Comment number 4 from reviewer 6 has not been addressed at all, despite the authors’

claim that it has been mentioned in Section 4.1.

Comment number 4 from reviewer 6: The details about parameters optimization is missing.

Response 6: After our experiments, using the Adam optimizer could make the model converge faster, with initial learning rates set to 0.0001, training cycles of 150, and batch sizes of 64. And the different fine-tuning strategies adopted for each model were listed in section 4.1

Point 7: Please list the six augmentation strategies in the revised manuscript.

Response 7: The six augmentation strategies were listed in the second paragraph of section 3.1.

Point 8: The explanation of “The FPS indicator represents the time taken by the model to detect an image or the number of images processed in one second.” is misleading. If we use time taken by the model to detect an image, then the smaller FTS will be a good indicator. However, if we use “the number of images processed in one second “ then a larger FPS value is will be a good indicator. This leads to contradictory values for a good indicator. Please clarify this point.

Response 8: The FPS indicator represents the number of images processed in one second.

We had corrected relevant descriptions in section 4.2.

Point 9: The response to why the choice of the 4 pretrained models in Reviewers 3 and 5 should address why these four were chosen over others, not addressing the issue of limited images.

Response 9: Transfer learning first required pre-training CNN on a large-scale ImageNet dataset. The four network structures selected in this paper, AlexNet, Vgg19, MobileNet V2, and ResNet50, were all based on CNN and had certain representativeness. From AlexNet to ResNet50, the number of layers in the network continued to increase. VGG19 increased the nonlinear ability of the network, MobileNet V2 accelerated the computational speed of the model, and ResNet50 added cross layer connections to solve the problem of gradient vanishing. The depth of the network layer could be adjusted by increasing the number of convolutional layers, pooling layers, or fully connected layers. These four networks compensated for their respective shortcomings and facilitated the extraction of rich and complex high-dimensional image features, making them suitable for the food-11 dataset. Therefore, these four models were selected as base learners.

We had corrected relevant descriptions in section 3.2.

---

## [Decision Letter · Decision Letter 2]

9 Oct 2023

PONE-D-23-15950R2Recognition of Food Images Based on Transfer Learning and Ensemble LearningPLOS ONE

Dear Dr. Hu,

Thank you for submitting your manuscript to PLOS ONE. After careful consideration, we feel that it has merit but does not fully meet PLOS ONE’s publication criteria as it currently stands. Therefore, we invite you to submit a revised version of the manuscript that addresses the points raised during the review process.

We look forward to receiving your revised manuscript.

Kind regards,

Nouman Ali

Academic Editor

PLOS ONE

Reviewers' comments:

Reviewer's Responses to Questions

**Comments to the Author**

1. If the authors have adequately addressed your comments raised in a previous round of review and you feel that this manuscript is now acceptable for publication, you may indicate that here to bypass the “Comments to the Author” section, enter your conflict of interest statement in the “Confidential to Editor” section, and submit your "Accept" recommendation.

Reviewer #5: All comments have been addressed

Reviewer #7: (No Response)

2. Is the manuscript technically sound, and do the data support the conclusions?

Reviewer #5: Yes

Reviewer #7: Yes

3. Has the statistical analysis been performed appropriately and rigorously? 

Reviewer #5: Yes

Reviewer #7: No

4. Have the authors made all data underlying the findings in their manuscript fully available?

Reviewer #5: Yes

Reviewer #7: Yes

5. Is the manuscript presented in an intelligible fashion and written in standard English?

Reviewer #5: Yes

Reviewer #7: No

6. Review Comments to the Author

Reviewer #5: (No Response)

Reviewer #7: The authors have not addressed my comments and probably they have upload the previous version again. I am re sending my comments again to them so that they can submit review carefully

The manuscript has already been revised and apparently, the quality seems to be good enough to be considered as a journal paper for review, keeping in view, my addition as a review for this manuscript at the stage of version R1, I will send the following queries to authors

1. In table 14, authors must cite the original research articles, currently only method name is there

2. A comparison in terms of a large-scale dataset is required.

3. Computational complexity is required to be discussed and presented in terms of Big-o-Notion.

4. How authors have selected research variables, it is not clear?

5. Manuscript must be proof-read to fix grammatical errors.

7. PLOS authors have the option to publish the peer review history of their article (what does this mean?). If published, this will include your full peer review and any attached files.

Reviewer #5: No

Reviewer #7: No

---

## [Author Response · Author response to Decision Letter 2]

18 Oct 2023

Original Manuscript ID: PONE-D-23-15950R1 

Original Article Title: “Recognition of Food Images Based on Transfer Learning and Ensemble Learning”

To: Editor

Re: Response to reviewers

Dear Editor,

Thanks for your letter and review work on our manuscript entitled “Recognition of Food Images Based on Transfer Learning and Ensemble Learning”. In the previous revision, we mistakenly believed that the questions in the attachment included all the questions from Reviewer 5 and Reviewer 7, so we only answered the questions in the attachment. We greatly appreciate the reviewers’ complimentary comments and suggestions. These comments are very helpful and valuable for improving and polishing our paper. The point-by-point responses to Reviewer 7 comments are given in the following part. 

Best regards,

Le Bu, Caiping Hu, Hui Sui, Xiuliang Zhang

 

Reviews and Responses

Point 1: In table 14, authors must cite the original research articles, currently only method name is there?

Response 1: There is no Table 14 in this article, I'm not sure if you're referring to Table 4. We temporarily understood it as the problem of Table 4. The citations of the original papers of different algorithms have been pointed out in the introduction part. The suffix "ft" indicated that this model underwent fine-tuning training under transfer learning methods, and different fine-tuning strategies have been pointed out in section 4.1. 

Point 2: A comparison in terms of a large-scale dataset is required.

Response 2: We compared it to the food-101 dataset in section 4.5.

Point 3: Computational complexity is required to be discussed and presented in terms of Big-o-Notion.

Response 3: We have added relevant descriptions in section 4.4.

Point 4: How authors have selected research variables, it is not clear?

Response 4: The main research variables in the paper were the parameters of the different models. We referred to the original papers of different models and determined the model parameters based on a large number of experiments and research experience.

Point 5: Manuscript must be proof-read to fix grammatical errors.

Response 5: We have carefully checked the grammar of the paper and ensured that there are no errors. If you detect any errors, please help me point them out. We will be very grateful.

---

## [Decision Letter · Decision Letter 3]

19 Dec 2023

Recognition of Food Images Based on Transfer Learning and Ensemble Learning

PONE-D-23-15950R3

Dear Dr. Hu,

We’re pleased to inform you that your manuscript has been judged scientifically suitable for publication and will be formally accepted for publication once it meets all outstanding technical requirements.

Kind regards,

Nouman Ali

Academic Editor

PLOS ONE

Additional Editor Comments (optional):

Reviewers' comments:

Reviewer's Responses to Questions

**Comments to the Author**

1. If the authors have adequately addressed your comments raised in a previous round of review and you feel that this manuscript is now acceptable for publication, you may indicate that here to bypass the “Comments to the Author” section, enter your conflict of interest statement in the “Confidential to Editor” section, and submit your "Accept" recommendation.

Reviewer #7: All comments have been addressed

2. Is the manuscript technically sound, and do the data support the conclusions?

Reviewer #7: Yes

3. Has the statistical analysis been performed appropriately and rigorously? 

Reviewer #7: Yes

4. Have the authors made all data underlying the findings in their manuscript fully available?

Reviewer #7: Yes

5. Is the manuscript presented in an intelligible fashion and written in standard English?

Reviewer #7: Yes

6. Review Comments to the Author

Reviewer #7: The authors have addressed my all suggestions and now this manuscript is up-to-the-mark as a journal paper

7. PLOS authors have the option to publish the peer review history of their article (what does this mean?). If published, this will include your full peer review and any attached files.

Reviewer #7: No

---

## [Editor Report · Acceptance letter]

9 Jan 2024

PONE-D-23-15950R3 

PLOS ONE

Dear Dr. Hu, 

I'm pleased to inform you that your manuscript has been deemed suitable for publication in PLOS ONE. Congratulations! Your manuscript is now being handed over to our production team.

Kind regards, 

on behalf of

Dr. Nouman Ali 

Academic Editor

PLOS ONE